# Tournament Evaluation of Large Language Models

## Abstract

For several decades, the standard approach to evaluating a learned model has been to compute a numerical loss that summarizes the quality of the model based on a previously unseen test set. Two models for the same task can then be compared by looking at their scores on this set. However, recent experience with large language models (LLMs) has shown that comparing summary statistics of two broadly-capable models may not provide a reliable predictor of performance on real-world tasks. This has led to a growing use of crowd-sourced human feedback directly comparing outputs from pairs of models. While helpful, this approach requires a process that involves significant time and human effort, limiting the number of models that can be thoroughly evaluated. To address the need for a scalable method of comparing modern LLMs, we present a novel approach to evaluation via tournament-style model competitions that are constructed automatically from pre-existing benchmarks. We use these automatically-constructed tournaments to compute ratings for a range of models on a diverse set of tasks that use automated scoring via both multiple-choice and free-form text generation. We compare four prominent rating systems: Elo, Glicko, TrueSkill™, and the Bradley-Terry model, and find that automatically-constructed tournaments provide reliable information about the relative performance of LLMs while using only a fraction of the amount of data required by current benchmark-based evaluation methods. We discuss implications for model evaluations and propose future directions for large-scale LLM comparisons.

## 1 Introduction

Machine learning models have historically been developed for narrowly defined *tasks*, which has enabled a simple and now well-understood procedure for evaluation: collect a large number of instances of the task, partition those instances into training and evaluation sets, find weights that minimize a loss function on the training set, and compute the loss on the previously unseen evaluation set. We declare models to have "learned" or "generalized" if performance on the unseen examples meets our requirements. Early definitions of machine learning go so far as to include the specification of a task in the definition of learning (Mitchell, 1997).

Recent developments have called into question the continuing utility of this approach: internet-scale datasets and the transformer architecture have led to the creation multi-billion parameter models that display broad capabilities in relation to language processing and vision. These large models have not necessarily been trained on a single task, making it hard to assess the extent to which the models generalize. In some cases it is not even possible to assess whether or not a potential instance of test data is truly "unseen" or "in distribution."

There have been two broad approaches to addressing this challenge: on the one hand, researchers have pursued the traditional approach, and have created a large number of benchmarks, with both training and test sets. These benchmarks have grown in size and difficulty as language models have improved. A new model is typically evaluated on several benchmarks, and its scores are either reported directly or aggregated into a single summary statistic. There are now several leaderboards that use benchmark performance to rank models.

At the same time, researchers have recognized the importance of direct comparison of language models, and have constructed systems that allow human comparison of the output of two language

models on a single task instance. This direct comparison data is used to compute scores using approaches such as Elo ratings that are updated over many comparisons. While helpful, this approach requires a process that involves significant time and human effort, limiting the number of models that can be thoroughly evaluated.

To address the need for a scalable method of comparing modern LLMs, we present a novel approach to evaluation via tournament-style model competitions that are constructed automatically from pre-existing benchmarks. We build off of the framework of the Evaluation Harness developed by EleutherAI (Gao et al., 2024). We use these automatically-constructed tournaments to compute ratings for a range of models on a diverse set of tasks that use automated scoring via both multiple-choice and free-form text generation. In this work we outline the details of this approach, and then describe our empirical evaluation of the method on a number of open models across a range of tasks. Our system is available as an open source project.[1]

## 2 RELATED WORK

Pairwise ranking systems have been used in a variety of LLM comparisons, using human feedback (Nichol et al., 2021; Askell et al., 2021; Bai et al., 2022), an LLM-as-judge (Dettmers et al., 2023; Zheng et al., 2023), or with synthetic data (Boubdir et al., 2023). The most popular is LMSYS' Chatbot Arena (Chiang et al., 2024), where users get to rank the output between two different models. Chatbot Arena's leaderboard initially used Elo but has been updated to use the Bradley-Terry model (with a similar initial ranking and scaling factor to Elo) (Chiang et al., 2023). They report the Bradley-Terry model to be favorable due to model performance being static and it producing better confidence intervals.

Boubdir et al. (2023) illustrates the properties of Elo as a ranking system between different LLMs, highlighting the importance of the hyperparameters in getting stable rankings. They use both synthetic data generated from a binomial distribution and human feedback to validate their experiments. Peyrard et al. (2021) utilizes the Bradley-Terry model, Elo, and TrueSkill to compare NLP systems, finding that Bradley-Terry disagrees with the mean and the median aggregation about 30% of the time about state-of-the-art models.

TrueSkill has been adopted in a couple of instances for NLP evaluation. Sakaguchi et al. (2014) adapt TrueSkill to rank machine translation based on human annotations. They note that the number of pairwise annotations needed to accurately rank models decreases when sampling the space of completions non-uniformly. Dušek et al. (2018) and Deriu et al. (2020) continue in this direction using TrueSkill, alongside a bootstrapping resampling technique, to cluster results for natural language generation and a Turing Test-like completion respectively. Chen et al. (2024) use TrueSkill in their LLMArena to assess performance in multiagent dynamic environments.

## 3 BACKGROUND

### 3.1 A BRIEF HISTORY OF METRICS FOR LANGUAGE MODELS

NLP has utilized various metrics to evaluate performance of language models since its inception. Early metrics for information retrieval focused on precision and recall or combined them into an F-measure (Manning et al., 2008; Jurafsky & Martin, 2024), while machine translation tasks often used bilingual human translators to evaluate performance. Papineni et al. (2002) introduced BLEU (Bilingual Evaluation Understudy), an automatic metric that evaluates machine translation quality by comparing the language model output to one or more human translations using n-gram overlap count. BLEU provided a quick, inexpensive, and language-independent metric that initially correlated well with human judgements, allowing for quicker development cycles for machine translation systems. Banerjee & Lavie (2005) introduced METEOR (Metric for Evaluation of Translation with Explicit ORdering) which improved upon BLEU by using heuristics to reward n-grams beyond exact matches, allowing for synonyms and paraphrases in the translation. Moving beyond n-gram metrics, Zhang et al. (2020) proposed BERTscore, which leverages contextual embeddings from a language model (i.e. BERT) to better capture semantic similarity between

---

[1]Link to be made available after anonymous review is complete.

reference texts.

Other automated metrics focus on Question and Answering, Rogers et al. (2023) and Cambazoglu et al. (2021) offer extensive overviews and taxonomies of these datasets. For rapid LLM development and hyperparameter tuning, Q&A datasets have become standard benchmarks. Human evaluation has regained popularity, with Askell et al. (2021) and Chiang et al. (2024), but continues to have much higher cost and can be noisy. The initial HuggingFace leaderboard used ARC (Clark et al., 2018), Hellaswag (Zellers et al., 2019), MMLU (Hendrycks et al., 2021a), TruthfulQA (Lin et al., 2022), Winograd (Sakaguchi et al., 2019), and GSM8k (Cobbe et al., 2021) for automated LLM evaluation. Due to rapid progression of LLM capacity, reaching near human level performance on many of the metrics, as well as potential data contamination in pretraining, the leaderboard has been updated (Fourrier et al., 2024). It now uses: IFEval (Zhou et al., 2023), BBH (Suzgun et al., 2022), MATH (Hendrycks et al., 2021b), GPQA (Rein et al., 2023), MuSR (Sprague et al., 2024), and MMLU-PRO (Wang et al., 2024).

## 4 Tournament Evaluation

We propose to use benchmark Q&A tasks designed for evaluation of a single model to construct matches that enable direct head-to-head comparisons between two models. To enable the comparison of multiple models, we organize sets of matches into tournaments that allow us to derive scores for each model of interest.

We define a *task* to be a triple $\mathcal{T} = (\mathcal{D}, F, \mu)$, where $\mathcal{D}$ is a set of *instances*, i.e. data points from a benchmark dataset, $F$ is a *filter function* (typically for string normalization and regex-based string processing), and $\mu$ is a *metric function*. In the standard approach to evaluation, a language model $L$ is applied to each instance $x_i \in \mathcal{D}$. The output $y_i = L(x_i)$ of the model is then passed through the filter $z_i = F(y_i)$, and the filtered output is used to compute a score via the metric function $m_i = \mu(z_i)$. The score $m_i$ for each instance $i$ is then aggregated into a single summary statistic that captures the performance of the model $L$ on the task $\mathcal{T}$.

**Hellaswag.** For a simple example, in the Hellaswag task of Zellers et al. (2019), each instance is a multiple choice question that presents an initial segment of text and four possible completions for that initial segment. One of the completions $c^\star$ is marked as the correct completion. In the usual way of evaluating a model on this task, the prefix $\Pi$ is concatenated with each of the completions $c_1, c_2, c_3, c_4$, and the probability of each concatenation is computed: $p_i = L((\Pi, c_i))$. In this case, the filter $F$ is trivial and $\mu$ compares the probabilities from the model with the correct completion:

$$m_i = \mu(p_1, p_2, p_3, p_4) = \begin{cases} 1 & \text{if } \arg\max_i\{p_i\} = c^\star \\ 0 & \text{otherwise} \end{cases} \tag{1}$$

The final score of $L$ on this task is then the mean of $\mu$ on all the instances in the dataset: $\frac{1}{|\mathcal{D}|} \sum_{i=1}^{|\mathcal{D}|} m_i$.

**GSM8k.** In the GSM8k task (Cobbe et al., 2021), the instances are strings containing grade school math word problems. The correct answers are given as strings containing a chain of reasoning, the string ####, and the final answer. In this case, the filter function $F$ performs string processing to extract the model's answer after #### and $\mu$ is 1 if and only if the extracted answer is exactly equal to the correct answer.

Given a task $\mathcal{T} = (\mathcal{D}, F, \mu)$, we define a *match* to consist of the task $\mathcal{T}$, a pair of models $L_1, L_2$, a match *schedule* defined as a sequence of indices $S = (i_1, \ldots, i_k)$, where each $i_j \leq |\mathcal{D}|$, and a comparison function $\kappa$ that determines, for a pair $(\mu(L_1(x)), \mu(L_2(x)))$, whether $L_1$ or $L_2$ has the better answer on instance $x$. In the case of a multiple choice task, we can define $\kappa$ as:

$$\kappa(L_1, L_2, x) = \begin{cases} (1, 0) & \text{if } \mu(L_1(x)) = 1 \text{ and } \mu(L_2(x)) = 0, \\ (0, 1) & \text{if } \mu(L_1(x)) = 0 \text{ and } \mu(L_2(x)) = 1, \\ (0, 0) & \text{if } \mu(L_1(x)) = \mu(L_2(x)). \end{cases} \tag{2}$$

The value of $\kappa$ can be interpreted as the number of *points* earned by each model on an instance. In the multiple choice context, a model $L_1$ gets a point when it answers a question correctly and its

oppenent $L_2$ does not. No points are awarded when both models are correct or incorrect. Using $\kappa$, we can define a point total $T_\alpha$ as

$$T_\alpha = \sum_{j=1}^{k} \pi_\alpha(\kappa(L_1, L_2, x_{i_j})), \qquad (3)$$

where $\pi_\alpha$ denotes projection onto coordinate $\alpha$, with $\alpha = 1, 2$. From these totals we can declare a winner of a match. Based on the winner of a match, we can update Elo scores for both models based on the procedure described in Section 5.1. One advantage of this approach is that if a task is too easy or too hard for both models being evaluated, then matches will result in a draw and neither model's rating will be impacted; only genuine differences in model performance are rewarded.

We can define a tournament as a collection of models $L_1, \ldots, L_M$, a collection of tasks $\mathcal{T}_1, \ldots, \mathcal{T}_K$, a number of *rounds* $N$ (the number of matches to run for each pair of models), a match size $k$, a rule for constructing match schedules, and a rule for determining which models are paired together in which order. Perhaps the simplest example of such a rule is *round-robin scheduling*, where each model is paired with every other model for $N$ matches of size $k$. For $M$ models, this leads to $M(M-1)/2$ matches, which may be prohibitively costly to run depending on the specific models chosen. One could then imagine other rules for pairing models, such as single-elimination matches or hybrid schedules.

## 5 RATING SYSTEMS

Let $f : \mathcal{M} \to \mathcal{R}$ be a function that maps a match to a rating, where: $\mathcal{M}$ is the set of all possible matches and $\mathcal{R}$ is the set of possible ratings. A rating system is defined as a tuple $(\mathcal{M}, \mathcal{R}, f, \mathcal{P}, \mathcal{U})$ where: $\mathcal{M}$ is a non-empty set of matches, $\mathcal{R} \subseteq \mathbb{R}$ is a non-empty set of possible ratings, $f : \mathcal{M} \to \mathcal{R}$ is the rating function, $\mathcal{P}$ is a non-empty set of participants, and $\mathcal{U} : \mathcal{P} \times \mathcal{M} \to \mathcal{R}$ is an update function that adjusts a participant's rating based on a match outcome. We require the conditions that $\forall m \in \mathcal{M}, \exists p_1, p_2 \in \mathcal{P}$, where $p_1 \neq p_2$, representing the participants in the match, and $\forall p \in \mathcal{P}, \forall m \in \mathcal{M}, \mathcal{U}(p, m) = r'$, where $r'$ is the updated rating for participant $p$ after match $m$.

### 5.1 ELO RATING SYSTEM

The Elo rating system is a method for calculating the relative skill levels in two-player competitions. Initially proposed by Arpad Elo in 1967 (Elo, 1967) for chess rankings, it has since been adapted for various sports and board games (Silver, 2014; Lezard, 2024). Elo calculates an expected score for a player, based on both their and their opponents current ratings. If player $A$ has a rating of $R_A$ and player $B$ has a rating of $R_B$ then the expected score for player $A$ is:

$$E_A = \frac{1}{1 + 10^{(R_B - R_A)/400}}. \qquad (4)$$

Due to the scaling factor above, a score difference of 400 points translates into a 10:1 odds in favor of the higher scored player. After each match, the player's rating is updated as follows:

$$\mathcal{U}(A, m_i) = R_A + K(S_A - E_A) = R_A'. \qquad (5)$$

$S_A$ is the outcome of the match, which for our tournaments can be 1, 0.5, or 0, for wins, draws, and losses respectively, and the $K$ scaling factor is a hyperparameter of the rating system. The choice of $K$ determines the maximal change in a player's rating after a single match. We investigate the effect of the $K$ factor in Section 6.1.4.

### 5.2 ALTERNATIVES TO ELO

#### 5.2.1 BRADLEY-TERRY MODEL

Originally formulated in 1952 by Ralph A. Bradley and Milton E. Terry (Bradley & Terry, 1952), the Bradley-Terry model is a probability model for pairwise comparison. Instead of a score, the

Bradley-Terry (BT) model assumes that each player has an underlying "strength" or "ability". When two players are compared, the probability of one winning is modeled as a function of their relative strengths. Specifically, for players $A$ and $B$, the probability that $A$ "beats" $B$ is given as:

$$Pr(A > B) = \frac{p_A}{p_A + p_B}, \tag{6}$$

where $p$ is a real-valued score assigned to both players. Assume we know the outcomes of a set of competitions between players, let $w_{AB}$ be the number of times player $A$ beats player $B$. The likelihood function is as follows:

$$L(p) = \prod_{A,B} \left( \frac{p_A}{p_A + p_B} \right)^{w_{AB}}. \tag{7}$$

### 5.2.2 GLICKO

The Glicko rating system, developed by Mark Glickman in 1995 (Glickman, 1995), offers an extension to the Elo rating system that addresses some of its limitations. The key innovation is to the Glicko system is the introduction of rating deviation, which measures the uncertainty in a players rating. Glicko-2 (Glickman, 2022) adds another parameter, rating volatility, which measures how consistent a player's performance is over time.

### 5.2.3 TRUESKILL

TrueSkill (Herbrich et al., 2007; Minka et al., 2018), developed and patented by Microsoft Research for Xbox Live, is a Bayesian skill rating system that generalizes both Elo and Glicko to support multi-player competitions. It models each player's skill as a Gaussian distribution with a mean $\mu$, the perceived skill, and a standard deviation $\sigma$, the uncertainty. TrueSkill employs factor graphs and message passing algorithms, specifically expectation propagation, to efficiently compute marginal skill distributions.

## 6 EMPIRICAL EVALUATION

We have conducted a series of tests to determine if the tournament-based evaluation method proposed above is suitable for practical evaluation of language models. Suitability can be defined based on several factors, but we focus broadly on two sets of criteria, related to what can be called internal consistency and benchmark consistency. *Internal consistency* refers to whether a set of ratings, considered on its own, gives a coherent snapshot of the performance of a set of models, while *benchmark consistency* refers to the extent to which tournaments (which may be derived from benchmark datasets using only a few samples) tend to be consistent with the results obtained by simply running benchmarks in the conventional manner. We also examine how the choice of rating system impacts tournament rankings and share some findings related to model quantization and benchmark saturation.

### 6.1 INTERNAL CONSISTENCY

Internal consistency is a baseline requirement for any model evaluation method. If possible, an evaluation should produce a total ordering on the evaluated models, and this ordering should be invariant under perturbations of the experimental design that produced the order. We first examine the extent to which tournament evaluations converge on stable ratings and rankings, and then examine how much the resulting rankings satisfy transitivity and invariance to experimental setup.

### 6.1.1 SCORE CONVERGENCE

In their most basic form, Elo updates can become unbounded in either the positive or negative direction depending on the number of tournaments run and the relative strength or weakness of the competitors. Like many others, we have found it useful to manually impose bounds on the Elo scores of the models under test. We use a hard floor of 100 for all models and a "soft ceiling" whereby updates for models with a rating above a threshold (chosen as 3000 in our experiments) are exponentially decayed in proportion to the difference between the model rating and the threshold.

We consider varying match size and the number of matches jointly. As an example, consider Figure 1, which shows the results of a sequence of tournaments between three variants of Meta's 8 billion parameter Llama-3.1: one that uses 16 bits per parameter, and two quantized models that use 8 and 4 bits. The picture is much clearer in this case. Up to a threshold number of total instances, the models remain close in score. However, there is a threshold past which model differences grow to show a clear differentiation in the models. As we should expect, model performance is increasing with bits per parameter. This is typical of our experience using tournament evaluation to compare quantized models; we have found that differences between quantization levels that may be difficult to detect via benchmarks can typically be made clear through the use of tournament evaluation.

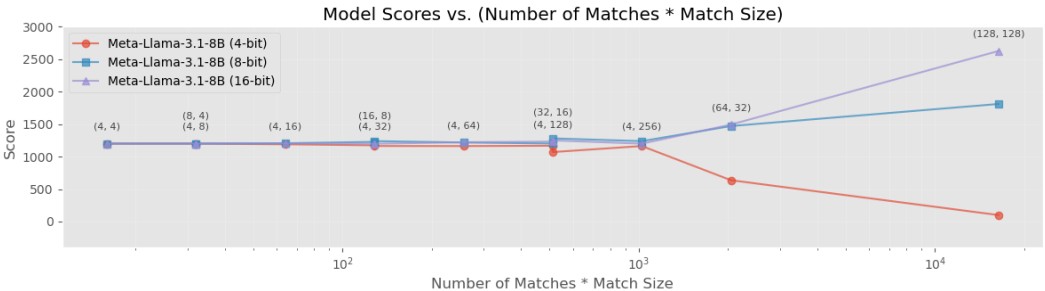

Figure 1: Performance of several quantized Llama models as total number of instances grows.

### 6.1.2 TRANSITIVITY

Previous work has shown that using Elo ratings with human evaluation can result in inaccurate rankings between models (Boubdir et al., 2023). That work observes that Elo ratings may fail to satisfy *transitivity* and *reliability*. To define transitivity, we introduce a relation $\preceq$: Given models $A$ and $B$, we write $A \preceq B$ to indicate that model $B$ is expected to beat model $A$. Given ratings $R_A$ and $R_B$, we expect at a minimum to have

$$A \preceq B \iff R_A < R_B. \tag{8}$$

Because ratings are real numbers, for any three models $A, B, C$, we always have a linear order on the ratings. If the ordering is $R_A < R_B < R_C$, we would expect then that $A \preceq B$ and $B \preceq C$. Transitivity would then hold if $A \preceq C$. However, Boubdir et al. (2023) show that in some instances we can have $R_A < R_B < R_C$, but $C \preceq A$. A rating system that satisfies transitivity allows us to compare models without having to run model-to-model comparisons for all pairs of models.

To explore the extent to which transitivity may hold for our tournament evaluation design, we chose a set of models, ran head-to-head tournaments for all pairs of models in our set, and then examined whether the resulting orderings, taken together, satisfy transitivity as described above. For evaluating transitivity, we selected seven widely-used open source models and ran them in head-to-head tournaments on a subset of the tasks used in version one of the popular Hugging Face LLM leaderboard (Fourrier et al., 2024). The tasks used were ARC (Clark et al., 2018), Hellaswag (Zellers et al., 2019), TruthfulQA Lin et al. (2022), Winogrande (Sakaguchi et al., 2019), and GSM8k (Cobbe et al., 2021).

Each head-to-head comparison was a single tournament between two models that consisted of 4 matches of 128 randomly-sampled instances from one of the tasks; this was done for each of the five tasks, so that each pair of models was evaluated on 2560 instances representing less than 20% of the total number of available instances. Updated Elo ratings were computed after each match. The models and a directed graph showing the order structure that resulted from the tournaments are displayed in Figure 2. That this results in a linear order is clear from the fact that all nodes are connected by at least one arc and the nodes can be topologically sorted in the linear order presented in Figure 2. We find that as long as enough instances are used to reach stable rankings among a set of models that transitivity failures do not appear to be a significant problem with tournament evaluation.

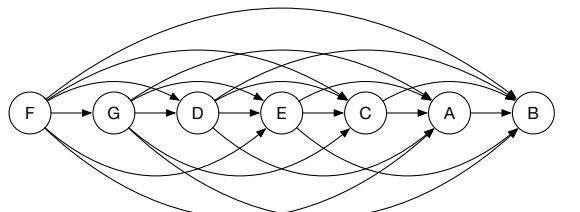

A   Qwen/Qwen2.5-7B-Instruct (8-bit quantized)
B   Qwen/Qwen2.5-7B-Instruct
C   microsoft/Phi-3.5-mini-instruct
D   meta-llama/Meta-Llama-3.1-8B-Instruct
E   meta-llama/Meta-Llama-3.1-8B-Instruct (8-bit quantized)
F   openai-community/gpt2
G   Qwen/Qwen2.5-0.5B-Instruct

Figure 2: Measurement of transitivity. An arc extends from node $A$ to node $B$ if the pairwise tournament described in Section 6.1.2 resulted in model $B$ receiving a higher Elo score than model $A$.

### 6.1.3 ORDER INVARIANCE

In addition to the requirement of transitivity, the work of Boubdir et al. (2023) discusses what they call "reliability," which encompasses two aspects: sensitivity to model ordering and hyperparameter sensitivity. We examine both of these aspects in our tournament setting, finding empirically that automatically-constructed tournaments are reliable in both senses. To test order invariance, we examined GPT-2 (137 million parameters) and Phi-3.5 (3.82 billion parameters). We ran two sets of tournaments in which each model competed against its 8- and 4-bit quantized variants on matches built from Hellaswag questions. We ran round-robin tournaments for all six permutations of model ordering and found in all cases that the resulting rankings were identical, and that the standard deviation in the resulting ratings was less than 2.95 averaged across the quantization levels.

### 6.1.4 HYPERPARAMETER SENSITIVITY

$K$ **Value.** The work of Boubdir et al. (2023) suggests that, in the case of Elo ratings, the choice of $K$ can make a significant difference in the behavior of the rating system. With this in mind, we ran a set of experiments with different $K$ values. As seen in Figure 3, if $K$ is too small, the scores are all clustered around their starting value of 1200 and if $K$ is too large, we see a saturation of values to the boundaries of 100 and 3000. All other experiments in this paper with Elo use a $K$ value of 10.

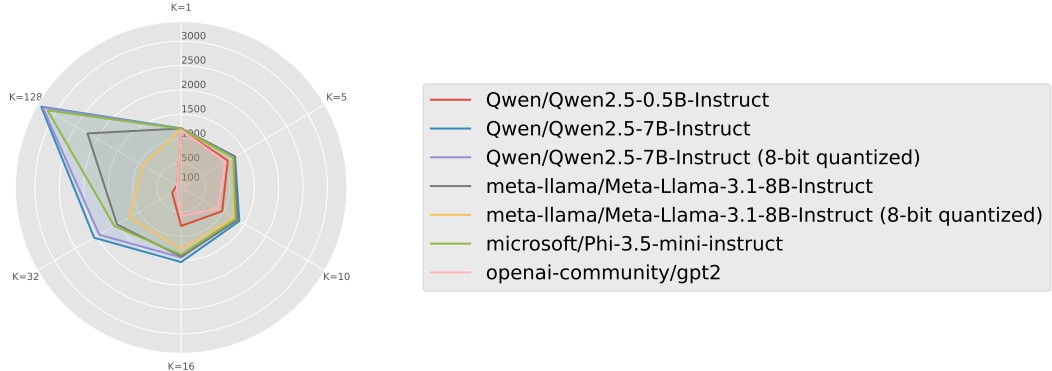

Figure 3: A radar chart showing the rating sensitivity to a choice of $K$ for the Elo update.

**Random Seed.** We also examined the extent to which the choice of random seed impacts our evaluations. While the choice of seed should not in principle matter, there is evidence from the computer vision literature that a judicious choice of random seed can distort the apparent performance of a system (Picard, 2021). We ran 8 tournaments using the Hellaswag dataset, each with a different random seed and then computed a series of statistics on the resulting Elo scores. The Shapiro-Wilk test (Shaprio & Wilk, 1965) produces a value of 0.002, demonstrating that the data is not normally distributed. Levene's test (Levene, 1961) for equal variances gives a value of 0.999, showing that the data has the same variance. Finally, we perform a one way ANOVA test, which gives an F-statistic

of 0.00 and a p-value of 1.00, allowing us to conclude that the random seed in fact does not impact the distribution of Elo scores.

## 6.2 BENCHMARK CONSISTENCY

Having shown that rankings derived from our dataset tournaments converge to sensible values and satisfy the requirement of consistency, we can ask whether or not the ratings and rankings determined via dataset tournaments are consistent with the ratings and rankings determined by widely-accepted evaluation procedures such as those used on the LLM Leaderboard (Fourrier et al., 2024).

We compare ratings and rankings for a set of models on a set of tasks taken from the Hugging Face LLM leaderboard with the ratings and rankings of those same models and tasks as determined by dataset tournaments. We use evaluation scores as computed by Hugging Face, and generate dataset tournament results using 4 rounds of match size 128. This allows us to compare the Elo scores of the models against the task ratings from the leaderboard.

We use the set of models listed in Figure 2, and the tasks from version 1 of the Open LLM Leaderboard, as described in Section 6.1.2. For each of those tasks, we run a round-robin tournament and record the resulting Elos. We then compare the resulting ratings and rankings against accuracies and rankings from the leaderboard, on both an individual per-task basis and averaged across all tasks. The resulting values for Pearson's correlation coefficient computed between accuracies and tournament scores and Spearman's rank correlation coefficient $\rho$ are shown in Table 1.

Table 1: Benchmark Consistency. For each task we compute the usual Pearson correlation coefficient between model ratings and accuracies, as well as Spearman's rank correlation coefficient $\rho$ as a measure of agreement between the rankings produced by accuracies and tournament ratings.

| Task | Pearson | Spearman $\rho$ |
|------|---------|-----------------|
| GSM8k | 0.946932 | 0.857143 |
| Hellaswag | 0.636327 | 0.815374 |
| Winogrande | 0.934180 | 0.964286 |
| TruthfulQA | 0.702241 | 0.571429 |
| Arc | 0.966429 | 0.892857 |
| mean | 0.944731 | 0.964286 |

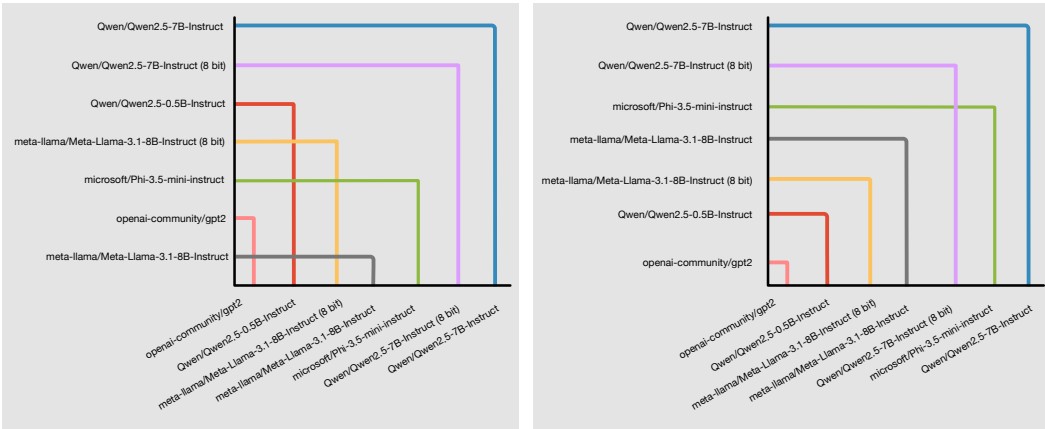

Figure 4: *Left:* Relationship between benchmark accuracy (horizontal) and tournament ratings (vertical) for the TruthfulQA task. *Right:* Same relationship, with accuracies and ratings averaged over all tasks. Horizontal axis orders models from lowest accuracy to highest accuracy. Vertical axis orders models from lowest tournament rating to highest.

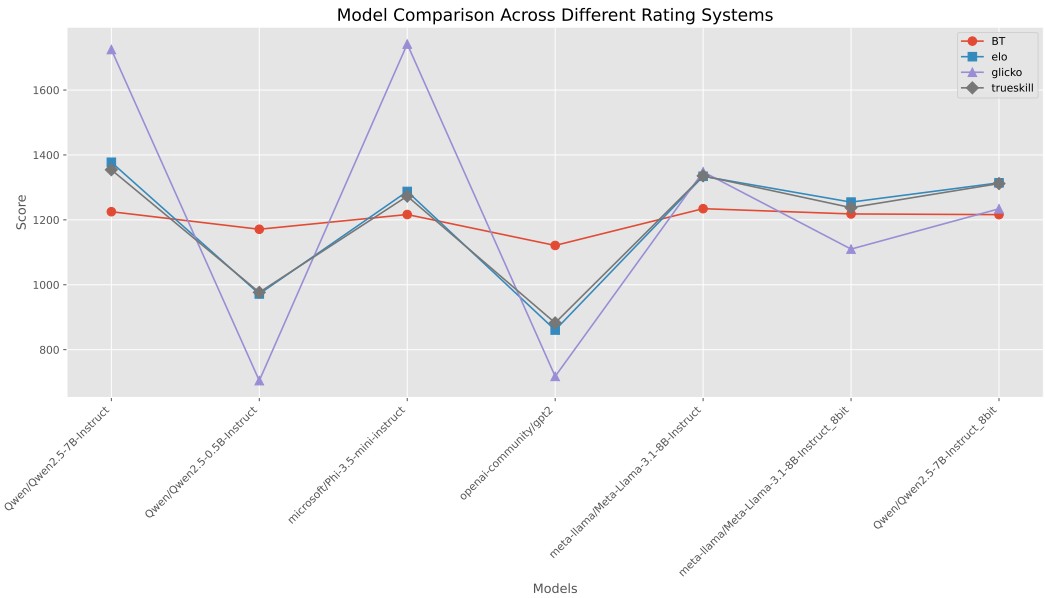

Figure 5: Rating Comparison between Bradley-Terry (BT), Elo, Glicko-2, and TrueSkill™.

We find that while correlation between benchmark accuracy and tournament performance is generally clear, there are some tasks where the correlation is weaker than expected.

In Figure 4, we show a "braid" diagram that shows how the rankings induced by benchmark accuracy relate to the rankings induced by tournament ratings. Each line crossing indicates a transposition that would be necessary to make the rating order equal to the accuracy order. We see that while the rating order for TruthfulQA by itself is not highly correlated with the accuracy order, averaging over multiple tasks shows increased correlation between benchmark accuracy and tournament ratings. We also note the surprisingly weak performance of Llama-3.1 on this task, which was consistently reflected in both accuracy and tournament performance; we plan to dedicate future effort to better understand why this happens.

### 6.3 COMPARING RATING SYSTEMS

We compare Elo to three other alternative ranking systems: Bradley-Terry, Glicko-2, and TrueSkill. With all three alternative rating systems we initialize the scores to be 1200 and attempt to scale the results to be in the same range as our Elo implementation. Our implementation of Glicko-2 follows (Glickman, 2022), initializing $\tau$ to 0.5, the rating deviation to 350, and volatility to 0.06. Keeping TrueSkill within similar ranges required a custom implementation, for the hyperparameters we set $\mu = 1200, \sigma = 400/3, \beta = 200, \tau = 5$ and used a draw probability of 0.1. For consistency, we implemented a hard floor of 100 and a soft ceiling of 3000 in all three alternative rating systems.

We ran a round-robin tournament for each rating system consisting of 4 rounds of match size 128, sampling instances from both ARC and Hellaswag. As seen in Figure 5, Glicko-2 has the most variance, while Bradley-Terry has the least. Our TrueSkill implementation tracks Elo very closely. These results warrant further research, which we discuss in Section 7.

## 7 DISCUSSION AND FUTURE WORK

There are a number of avenues to consider for possible future work. Our work has been largely empirical thus far. We suspect that a theoretical analysis would provide insight into when tournament-based evaluation is most useful, and may provide guarantees of asymptotic behavior and bounds on rating differences based on benchmark performance. One of the practical challenges we have encountered in our evaluations is knowing how to set the "tournament hyperparameters" of the number

of matches per tournament and the match size. Additional empirical or theoretical guidance in this area would be helpful to develop.

Relatedly, our initial work here has shown that the choice of rating system can make a difference in how models are ultimately ranked against each other. This, along with the fact that more sophisticated rating systems offer additional information suggests that better characterizing rating system performance merits additional investigation. This will particularly be the case for tournament structures that differ significantly from the round-robin approach we have used in the present work. Factors such as Glicko's rating deviation will be particularly useful in tournament structures that involve elimination, where not all models will run the same number of times. Working out the details of how this impacts evaluations on a larger scale will be interesting future

Our evaluation software currently tracks the performance of every model on every instance that is used in an evaluation. For each such instance, the rating of the model at the time of evaluation is recorded. This means we can compute an average rating of models that do well on an instance, opening the door to more sophisticated sampling strategies. In particular, we are exploring how to create "synthetic tasks" by sampling from a set of otherwise unrelated data sets based on a target range of Elo ratings. This will allow us to construct tasks that are tailored to the capabilities of the models under test, which should be more informative than tasks chosen at random. The feedback from models back to data sets may also open other lines of research that we have not yet fully considered.

## 8 CONCLUSION

Evaluating language models and other transformer-based neural network systems will likely remain challenging for the foreseeable future, while we work to understand the full extent of such broadly capable models. However, the use of head-to-head comparison of models in the form of our proposed tournament approach offers a method to easily and automatically compare models with as much or as little data and compute as may be available. We hope that our approach and related software can simplify the seemingly insurmountable challenge of deciding which of a set of capable models should actually be used in practical applications.

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
