# OpenReview forum: "Tournament Evaluation of Large Language Models"
_ICLR.cc/2025/Conference — Submitted to ICLR 2025_

### Official Review · Reviewer_zKEr · 2024-10-26

**Soundness:** 1
**Presentation:** 2
**Contribution:** 1
**Rating:** 3
**Confidence:** 4

**Summary:**

This paper proposes tournament evaluation for evaluating LLMs. A tournament is composed of several LLMs to be compared, some tasks (benchmark datasets), and the number of instances in each dataset to be used, and the number of *match*, which compares two models based on some data instances. The relative strength of LLMs can be calculated by rating systems including Elo, Bradley-Terry model, Glicko, and TrueSkill. This paper conducts empirical evaluation of several LLMs using tournament evaluation and show that the results is consistent and the results of the tournament evaluation has transitivity.

**Strengths:**

- This paper proposes an interesting idea of comparing different LLMs using tournament evaluation.
- This paper brings many alternative rating systems to the field of LLM evaluation that are currently not widely used, including Glicko and TrueSkill.

**Weaknesses:**

- The paper does not provide thorough and sound experiments to justify the effectiveness of tournament evaluation. Detailed weaknesses
    - The paper only uses 7 *LLMs* for comparison, while one of the LLMs this paper compares is a 137M GPT2. As a paper that proposes a new benchmarking method, more comprehensive results on more LLMs should be provided.
    - The rationales of the selected tasks (old open LLM benchmark tasks) are not well justified. In Section 3.1, the paper mentions some issues of the old version of the open LLM benchmark. However, the results of this paper are mostly based on the saturated benchmark.
- The contribution of this paper is not very clear. The paper does not explain why tournament explanations are better than simply aggregating the scores/performance of several benchmarks of each LLM. It is also unclear why the proposed method should be consistent with the ranking of benchmark (benchmark consistency defined in the paper). If benchmark consistency is what we want, why don’t we just use the benchmark to compare LLMs?
- The experiment setting is somewhat unclear. For example, the experiment setting corresponding to Figure 1 and Section 6.1.1 is never specified.
- The notations and framework are complex and not easy to understand. It would be better to have a figure or table to clearly illustrate all the terms and notations used in the framework and their relations. The product notation in Equation (7) is unclear. The $\pi_\alpha$ in Line 168 on page 4 is unclear.
- The takeaway from the results of comparing different rating systems is unclear. More in-depth discussions and experiments are required.
- The introduction section’s bibliography is far from satisfactory. This section only cites two papers. However, the section is filled with prior works that are not properly cited and unsupported claims that should be supported by prior works.

**Questions:**

- What does the random seed in Section 6.1.4 affect?
- The sentence on page 8, Line 388, is very hard to understand. The sentence on page 6 Line 322 is odd and ungramatical

---

### Official Review · Reviewer_j392 · 2024-10-31

**Soundness:** 2
**Presentation:** 3
**Contribution:** 2
**Rating:** 5
**Confidence:** 4

**Summary:**

The main contribution of the paper is a detailed analysis of tournament style of evaluation for LLMs. The authors perform evaluation across MCQ and free-form text generation using 4 rating systems (Elo, Glicko, TrueSkill, Bradley-Terry).

**Strengths:**

- The findings of this work are quite interesting and would be very useful in a practical sense.

**Weaknesses:**

- The paper claims to put forth a novel method of evaluation; however, that contribution is very obscure. I urge the authors to highlight the exact details in the paper and, if possible, condense it as a separate algorithm block for clear demonstration. Additionally, a comparison of the proposed evaluation algorithm to some baselines would serve as good visualization and proof of concept. Specifically:
    - What is the exact algorithm being proposed, and how does it differ from existing evaluation methodologies? I assume that one of the fundamental baselines would be a brute-force comparison with a rating system, which would lead to exponential comparisons for a large number of models and datasets.
    - How is the proposed evaluation algorithm better than existing methodologies? By extension, what are the existing evaluation methodologies over which the novelty is claimed? I feel these should be documented in the Related Works (RW). I see that Section 6 discusses metrics to objectively assess the proposed evaluation strategy. How do other baselines perform with these metrics?

- The flow between sections is digressive and confusing. Section 5 on `Rating Systems` reads like a survey, and it would be better placed in Related Works, which discusses previous works in detail as a quick recap. Section 4 proposes the idea of tournament-style evaluation (Methodology), and Section 6 should follow immediately as Results and Discussions. Even within Section 4, the definitions of `HellaSwag` and `GSM8k` could be moved to the end, as they disrupt the flow. Similarly, Section 6.1.4 is disconnected from preceding sections. While it’s discussed as an extension of 6.1.3, it is unrelated to tournament-style evaluation, focusing instead on parameter tuning for Elo. It belongs with Section 5.1 and is not well-suited to Section 6, which concerns the results and discussion of the proposed (tournament-style) evaluation method.

- In its current state, the paper requires a complete rewrite to create a more cohesive flow without digressions and breaks. The introduction and abstract also need content that highlights precisely why the proposed method is superior, in what ways, and how the standard evaluation setup suffers without it. In summary, the introduction does not clearly motivate the upcoming contribution.

**Questions:**

- What is `mean` in Table-1? I am guessing it is not a mean across the various tasks.
- How is the set $\mathcal{S}$ of indices selected from dataset $\mathcal{D}$? Is it a random sample? Are the same number of samples selected from each dataset? Additionally, how are the models paired up for evaluation—is there a set rule for selection?
- In Section 6.1.5, what random value is the random seed controlling?
- If possible, a quick flowchart or algorithm block representation of the evaluation strategy would be helpful.
- The first paragraph of Section 5 seems unnecessarily convoluted with mathematical formulation, and those definitions are not useful beyond that paragraph. Please try to simplify it, or provide one straightforward definition.
- Elo can get quite tedious with multiple models, so a "sparse-Elo" should also be explored, where Elo is run in a dense fashion ($N \choose 2$ comparisons) for a few rounds, and then a model is only compared with models in a certain bracket around it, say R-1000 to R+1000, where R is the rating of the current model.
- Lines 119-121: Providing the full forms of those benchmarks, along with a brief description of the task being evaluated, would be beneficial for first-time readers and aid in quick reference.

---

### Official Review · Reviewer_aACx · 2024-11-02

**Soundness:** 1
**Presentation:** 4
**Contribution:** 2
**Rating:** 3
**Confidence:** 3

**Summary:**

This paper proposes a new LLM evaluation paradigm that constructs head-to-head tournaments from existing benchmarks. The authors claim that evaluating models in this way reliably captures the relative performance of LLMs while using only a small portion of the dataset when compared to running these benchmarks on each model separately. The work also addresses common problems with elo-based ranking systems and empirically demonstrates the method's robustness to them. The paper also includes an important discussion of the future work in this direction and the importance of having a scalable method for comparing LLMs.

**Strengths:**

The ideas of this paper are incredibly interesting. The authors describe the problem eloquently and their proposed solution is creative and novel. The presentation of the paper is also very pleasant, and it is, for the most part, very easy to follow and read. They lay down the groundwork of this idea in a clear manner and are extensive in their references and explanations of prior work. The limitations of the method are also discussed thoroughly and a valuable discussion about future work is provided.

**Weaknesses:**

My only issue with this paper is that I think the experiments do not seem to substantiate the claims. While there are experiments that I found to be interesting and valuable (namely the transitivity and random seed experiments/analyses), I have some concerns regarding a few of them. My concerns are expanded on in the questions part of this review.

Furthermore, I would have loved to see some more of the results of their experiments. The work very clearly had a lot of experiments done yet the ones presented are mostly only aggregates. I think the inclusion of an appendix with all the scores generated from every experiment would be very valuable for deeper analysis and clarity for the readers.

**Questions:**

The following are the parts of this paper that I found to be unsubstantiated. Any of the following could be due to a misunderstanding; I am willing to discuss this further.
## Order invariance inconclusivity
The Boubdir et al. [1] paper does this experiment at a much larger scale, which could very well be the reason that the issue of order variance emerges. Furthermore, the quantized versions of Phi and GPT-2 could have a significant performance gap, which could cause the order to matter less. Boubdir et al. find that this problem is likely exacerbated when models are similar in performance. It would be interesting to explore whether a larger-scale order invariance test could yield different insights.

## K-value
Why choose a K value of 10? There seems to be a switch in ranking for meta-llama/Meta-Llama-3.1-8B-Instruct starting from 16. It would be helpful to understand the rationale behind the choice of K=10, especially given the observed ranking differences when K=32. Could additional insight into this choice be provided? I would love to know your opinion on this.

## Data/compute quantity
You mention in the conclusion: "However, the use of head-to-head comparison of models in the form of our proposed tournament approach offers a method to easily and automatically compare models with as much or as little data and compute as may be available".

In section 6.2, you make the argument that there is a correlation between benchmark accuracy and tournament performance averaged over the tasks.
1. How are the tasks averaged? I'm not sure I fully understand the process here
2. The compute and data used for this experiment seem to be quite substantial. If I understand correctly, a tournament of 4 rounds with 128 tasks would have 84 total matches, with each running 2 * 128 = 256 forward passes. So this means there is a total of 21504 forward passes. When dividing that by the number of models 7, we get a total of 3072 forward passes per model, which is a lot more compute-intensive than running the entirety of the TruthfulQA dataset of 817 samples for each model. It seems to me that this doesn't support the claim that the method works "with as much or as little data and compute as may be available," especially since it was shown in Figure 1 that a smaller (number of matches * match size) results in scores that don't conclusively convey a reliable performance ranking. Please correct me if I misunderstood something and miscalculated.
## Quantization differences
"This is typical of our experience using tournament evaluation to compare quantized models; we have found that differences between quantization levels that may be difficult to detect via benchmarks can typically be made clear through the use of tournament evaluation."
- What dataset is being used for Figure 1?
- What results do you mean by "This is typical of our experience"? I think the results of the other quantized/non-quantized pairs tested should be presented to support this claim.

### Minor grammar issue
232 in a players should be in a player's

------------
[1] https://arxiv.org/pdf/2311.17295

---

### Official Review · Reviewer_Ertc · 2024-11-04

**Soundness:** 2
**Presentation:** 2
**Contribution:** 2
**Rating:** 3
**Confidence:** 4

**Summary:**

This paper adapts existing automated benchmarks for model pairwise comparisons within a scalable tournament-like setting, offering an alternative to human preference evaluations. t introduces an evaluation metric κ that depends on the task evaluated, using either exact match for straightforward answers or probability comparison for selecting the most likely completion from multiple choices. The paper tests the evaluation method for transitivity, order invariance, and  K-factor sensitivity.  Then, it assesses how rankings derived from benchmarks correlate with those produced by their tournament-based approach. Lastly, the work compares scores computed by popular rating systems like Elo and TrueSkill.

**Strengths:**

A comprehensive comparison of Elo with other rating systems such as Glicko, Trueskill and Bradley-Terry, which is lacking in the literature of LLMs evaluation.

**Weaknesses:**

- Missing literature citations in the introduction and lack of distinct structure between the "Related Work" and "Background" sections, which could be improved by either merging them into a single cohesive section or by defining a clearer separation of the topics discussed within each.

- Limited evidence that the tournament method offers deeper or new insights into model capabilities compared to traditional accuracy-based benchmarks, as it reuses these data point. The scalability advantage argument does also apply to conventional benchmarking.

- The work assumes properties such as transitivity or order invariance without stress-testing the tournament setting under conditions that could challenge these properties, such as models with very close win rates. The sensitivity of the Elo rating system to hyperparameters like the K-factor and its potential volatility, as shown in the literature,  in closely matches scenarios are not sufficiently addressed, which could question the reliability and stability of the evaluation outcomes under different settings.

- Initial data used for computing the scores is not provided, such as accuracy on benchmarks and win rates (κ values).

**Questions:**

1. Figure 1 shows the performance of quantized Llama models as the total number of evaluated instances grows, where each match consists of multiple instances. Typically, in Elo rating systems, a match would include only a single comparison between two models. Can the authors clarify their decision to include multiple instances per match? Furthermore, the plot aims to show how Elo scores diverge with increasing match size * number of matches. Averaging the Elo scores over multiple instance orderings (N reorderings) provide more stable and accurate results reflectove of actual win rates. I assume that no averaging over different "matches" reorderings  has been considered when computing the Elo scores, as you only average within each match (k index in equation 3).
2. Regarding the transitivity results, the Elo rating system shows failure at specific scenarios where win rates closely match (i.e. 0.49 vs. 0.51). In case the data used here only includes win rates that are skewed  (i.e. 0.65 vs. 0.35), it is difficult to assume transitivity of the tournament setting. Could you provide these rates for reference for all experiments discussed in the paper?

---

### Comment · Area_Chair_T24b · 2024-11-21
**Reminder: Please respond and update the score if necessary**

Dear Reviewers,

Kindly ensure that you respond proactively to the authors' replies (once they are available) so we can foster a productive discussion. If necessary, please update your score accordingly. We greatly appreciate the time and effort you’ve dedicated to the review process, and your contributions are key to making this process run smoothly.

Thank you,

AC

---

### Meta-Review · Area_Chair_T24b · 2024-12-13

**Metareview:**

This paper provides a detailed comparison of the Elo rating system with other systems such as Glicko, Trueskill, and Bradley-Terry, identifying a significant gap in the literature related to LLM evaluation. The authors propose an innovative LLM evaluation method that employs head-to-head tournaments using existing benchmarks. This approach efficiently captures the relative performance of LLMs while utilizing only a fraction of the dataset, unlike traditional methods that evaluate each model on complete benchmarks. The paper addresses common challenges associated with Elo-based rankings, showcasing the robustness of their method, and underscores the need for scalable approaches to LLM comparison, offering insights for future research in this domain.

However, all reviewers agree that the paper does not meet ICLR standards, particularly in terms of clarity, as highlighted by Reviewer j392. The paper is difficult to comprehend, and Reviewer aACx notes that the experiments do not adequately support the claims. Additionally, Reviewer Ertc points out the absence of a thorough literature review and relevant citations. Due to these significant shortcomings, I recommend rejecting the paper.

**Additional Comments On Reviewer Discussion:**

No changes or developments have been observed, as neither the reviewers nor the authors engaged in any follow-up discussions. I have encouraged the reviewers and authors to start the discussion during the author response period.

---

### Decision · Program_Chairs · 2025-01-22

Reject